# Two-Sided Antibacterial Cellulose Combining Probiotics and Silver Nanoparticles

**DOI:** 10.3390/molecules26102848

**Published:** 2021-05-11

**Authors:** Laura Sabio, Andrea Sosa, José M. Delgado-López, José M. Dominguez-Vera

**Affiliations:** Departamento de Química Inorgánica, Universidad de Granada, 18071 Granada, Spain; laurasabio@ugr.es (L.S.); sosa@correo.ugr.es (A.S.); jmdl@ugr.es (J.M.D.-L.)

**Keywords:** bacterial cellulose, Ag nanoparticles, probiotics, antibiotic-resistant bacteria

## Abstract

The constant increase of antibiotic-resistant bacteria demands the design of novel antibiotic-free materials. The combination of antibacterials in a biocompatible biomaterial is a very promising strategy to treat infections caused by a broader spectrum of resistant pathogens. Here, we combined two antibacterials, silver nanoparticles (AgNPs) and living probiotics (*Lactobacillus fermentum*, *Lf*), using bacterial cellulose (BC) as scaffold. By controlling the loading of each antibacterial at opposite BC sides, we obtained a two-sided biomaterial (AgNP-BC-*Lf*) with a high density of alive and metabolically active probiotics on one surface and AgNPs on the opposite one, being probiotics well preserved from the killer effect of AgNPs. The resulting two-sided biomaterial was characterized by Field-Emission Scanning Electron Microscopy (FESEM) and Confocal Laser Scanning Microscopy (CLSM). The antibacterial capacity against *Pseudomonas aeruginosa* (*PA*), an opportunistic pathogen responsible for a broad range of skin infections, was also assessed by agar diffusion tests in pathogen-favorable media. Results showed an enhanced activity against *PA* when both antibacterials were combined into BC (AgNP-BC-*Lf*) with respect to BC containing only one of the antibacterials, BC-*Lf* or AgNP-BC. Therefore, AgNP-BC-*Lf* is an antibiotic-free biomaterial that can be useful for the therapy of topical bacterial infections.

## 1. Main

The dramatic increase of antibiotic-resistant bacteria is one of the biggest threats to global health [1]. The development of innovative antibiotic-free antibacterials is, in fact, one of the most important challenges of material scientists [2,3]. Two promising alternatives to antibiotics are the use of silver nanoparticles (AgNPs) and probiotics.

AgNPs have received much attention as antibacterial agents [4,5,6,7,8]. Indeed, they are currently being used as antibacterials in medical devices, textiles, cosmetics, and food packaging [9,10,11,12]. AgNPs present antimicrobial activity as a result of the slow oxidation of Ag^0^ to Ag^+^ by air oxygen. Ag^+^ interacts with cys-rich proteins and perturbs their functionality, provokes RNA and DNA damages, and induces the production of highly toxic reactive oxygen species (ROS) [8,13]. Nevertheless, it should be noted that some strains have developed resistance to AgNPs [14].

Probiotics are live microorganisms known to provide health benefits to the host, either by restoring the natural balance of bacteria in the microbiota [15], by excreting antipathogenic compounds [16] or by restoring the pH of infected media to healthy values, as it occurs, for instance, in the therapy of bacterial vaginosis [17].

Each antibacterial usually covers a specific group of microbes. Combining several antibacterial agents in a single biomaterial has, therefore, the advantage to broaden the activity spectrum, increase its efficiency and be effective against complex bacterial infections in which more than one pathogen is involved.

Taking all these aspects into account, we have explored the possibility of combining living probiotics and AgNPs in a matrix as bacterial cellulose (BC) to produce a new hybrid biomaterial with enhanced antibacterial properties. In principle, this approach has the drawback that probiotic bacteria, in particular *Lactobacilli*, are highly susceptible to AgNPs [18]. To overcome this issue, we have developed a two-sided material, in which AgNPs and the probiotic *Lactobacillus fermentum* (*Lf*) are located on opposite BC faces, thus *Lf* being protected from the antibacterial AgNPs. In addition, the acidic environment caused by the excretion of lactic acid by *Lf* promotes Ag dissolution and ROS overproduction, thus increasing the bactericidal effect of AgNPs [18].

BC is a polymer of glucose made by some aerobic bacteria that has been widely studied for biomedical applications and, in particular, as a wound-dressing material [19,20,21]. BC is chemically pure, biocompatible, and due to its higher surface area, it exhibits extraordinary efficacy to absorb wound exudates without adhering to the wound surface, thereby avoiding tissue damage upon removal.

However, BC itself has no activity against bacterial infection, which is a recurrent issue affecting chronic or acute wounds. Most of the strategies used to confer antibacterial properties to BC are based in the covalent or non-covalent incorporation of antibacterial polymers and peptides to the glucose fibber network [22,23]. However, the chemical modification by covalent attachment of molecules to BC is difficult due to the poor solubility of BC, which makes necessary the employ of organic solvents, elevated temperatures, and long reaction times, limiting its large-scale production. The non-covalent incorporation of active biomolecules is chemically easier, but due to the weak interaction between BC and these biomolecules, the resulting biomaterial potentially suffers from shedding.

A strategy to produce antibacterial BC-derivatives that circumvents the chemical functionalization of BC and does not exhibit shedding has been the incorporation of Ag and TiO_2_ nanoparticles into BC [24,25]. Roig-Sanchez et al. have recently reported that, in fact, layers of BC can be decorated step-by-step with different nanoparticles (Ag and TiO_2_, among others), thus creating a *mille-feuille* concept of multifunctional BC [24]. On the other hand, we have recently demonstrated that BC can incorporate a huge number of probiotics resulting in a living biomaterial with enhanced antibacterial activity [26].

Taking advantages of these possibilities of BC, we have developed a BC derivative that combines two antibacterials with different mechanisms of action, AgNPs and living probiotics, *Lf*. We have used the step-by-step loading to host both antibacterials at opposite BC sides, thus obtaining a two-sided biomaterial (AgNP-BC-*Lf*) with a high density of alive and metabolically active probiotics on one surface and AgNPs on the opposite one, being probiotics preserved from the antibacterial AgNPs. No example of this type of antibacterial biomaterial has been reported so far.

*Acetobacter xylinum (Ax)* was used to obtain BC pellicles. Subsequently, probiotics and AgNPs were sequentially adsorbed on opposite faces of BC. First, one of the surfaces of BC was immersed in a probiotic culture (Step 1, Figure 1A). The adsorption of *Lf* on BC occurred very fast, and after a few hours, the number of loaded probiotics practically did not increase with incubation time. The resulting sample (BC-*Lf*) was repeatedly washed and then stained with the standard SYTO9/propidium iodide (PI) dyes to study the viability (live/dead) and allocation of probiotics within BC by confocal laser scanning microscopy (CLSM). As illustrated in Figure 2A,D, *Lf* only penetrated a few layers of BC. The dense fibril network does not allow the penetration of probiotics into the entire BC. The high density of live probiotics (green spots), points out that the transfer from the culture to the cellulose did not affect the probiotic viability.

In a second step, the opposite surface, free of probiotics, was functionalized with AgNPs (Step 2, Figure 1A). AgNPs with average diameter of 50 nm (Figure 1B,C) were synthesized by the Turkevich method, using citrate as a reducing and capping agent [27]. A suite of materials with different contents of AgNPs was obtained by incubating BC-*Lf* in the AgNPs-containing solution at different times, ranging from 5 to 30 min. The resulting materials were stained with SYTO9/PI and visualized by CLSM (Figure 2). We found that the longer the incubation time, the higher the number of dead probiotics (red spots) in the opposite face to that in contact with AgNPs (Figure 2B,C). This result was expected since, while probiotics cannot diffuse through the BC network, AgNPs (and Ag^+^ ions) diffuse through the entire BC structure, contacting the probiotics and affecting their viability. In any case, the number of no viable probiotics (red spots) was always significantly lower than that of alive ones (green spots) in the time interval here explored (5–30 min).

The biomaterial containing the highest AgNPs concentration with null effect on the probiotic viability was obtained by incubating BC-*Lf* for 15 min in the AgNPs solution (Figure 2B,E). The incorporation level of *Lf* in the final biomaterial, AgNP-BC-*Lf*, was determined after sample digestion with cellulase (see Materials and methods), by counting CFUs in MRS-agar plates after 24 h of incubation (37 °C) in anaerobic conditions. A value of 2 × 10^9^ CFUs/g (±8%) was obtained.

Field-emission scanning electron microscopy (FESEM) images of AgNP-BC-*Lf* confirmed that, as expected, it consists of a two-sided material: one face showing a high density of *Lf* with its typical morphology (Figure 3A), while the opposite face contained AgNPs, entrapped in the BC fiber network (Figure 3B). It is interesting to note that FESEM images of AgNP-BC-*Lf* showed neither AgNPs on the probiotic-containing face nor probiotic on the AgNPs-containing one. This material was further used to study the antibacterial activity (vide infra). In contrast, the concomitant presence of probiotics and AgNPs was observed at the probiotic-containing side of the sample obtained at longer incubation times (>15 min) (Figure 3C).

The antibacterial activity of the two AgNP-BC-*Lf* surfaces, containing either probiotics or AgNPs, was assessed against *PA,* an opportunistic pathogen responsible of a broad range of skin infections. The activities were compared to those of the materials containing only one of the antibacterials (AgNP-BC or BC-*Lf*). As expected, BC exhibited no activity against *PA*, while AgNP-BC-*Lf,* AgNP-BC, and BC-*Lf* materials produced clear inhibition zones against this pathogen (Figure 4). Importantly, the bifunctional AgNP-BC-*Lf* was more active against *PA* than BC-*Lf* or AgNP-BC, independently of the face exposed to the pathogen (**AgNP**-BC-*Lf* or AgNP-BC-***Lf***, Figure 4). These results confirm the existence of an additive effect between the two antimicrobial faces.

To understand this additive effect, it is interesting to note that the activity of the *Lf*-side of BC-*Lf* against *PA* (BC-***Lf ***in Figure 4) was only slightly higher than that of the probiotic-free face (**BC**-*Lf* in Figure 4). This finding suggests that the activity of BC-*Lf* is due to excreted species that diffuse through the BC structure and reach the pathogen media.

On the other hand, it is interesting to note that the antibacterial activity of AgNP-BC-*Lf*, no matter its face, is really a sum of the two antibacterial components (AgNPs and *Lf*) (Table 1). The possible silver cations diffusion inside the biomaterial during the agar-diffusion test, from AgNPs to *Lf*, and the consequent probiotic death, resulted low. In fact, the probiotic viability before and after the antibacterial test assays was close, with CFUs decreasing lower than 10% (see Materials and Methods).

## 2. Materials and Methods

### 2.1. Reagents and Solutions

High-grade quality reagents were used as received from commercial suppliers. Aqueous solutions were prepared with ultrapure water (18.2 MΩ cm, Bacteria < 0.1 CFU/mL at 25 °C, Milli-Q, Millipore, Burlington, MA, USA).

### 2.2. Synthesis of Silver Nanoparticles

Silver nanoparticles (AgNPs) were prepared by the well-known Turkevich method, using citrate as a reducing and capping agent [27]. Briefly, 10 mL of a 1 mM AgNO_3_ (Sigma) solution in deionized water was heated until it started to boil. After that, 10 mL of an aqueous solution of 5 mM sodium citrate (Sigma) was dropwise to the silver nitrate solution. The heating was continued for 10 min, and then cooled to room temperature. The solution was filtered using a 0.22 µm Sartorius filter prior to use. An absorption spectrum was recorded with the Unicam UV 300 spectrophotometer to observe the plasmon absorbance of Ag colloids.

Diameter of the particles was determined by electron microscopy. High-resolution Transmission electron microscopy (HR-TEM) images were recorded with a 300 kV FEI TITAN G2 60–300 microscope (Thermo Fisher Scientific, Waltham, MA, USA) of the Centre for Scientific Instrumentation, University of Granada (CIC-UGR, Granada, Spain). AgNPs were also imaged by Scanning transmission electron microscopy (STEM) using a high-angle annular dark field (HAADF) detector. AgNPs diameter distribution (histogram) was estimated by measuring the diameter of 100 nanoparticles with ImageJ software (version 1.48v; NIH, Bethesda, MD, USA).

### 2.3. Bacteria Culture

Lyophilized *Pseudomonas aeruginosa, PA* (ATCC 27853, CECT 108), and *Acetobacter xylinum, Ax* (ATCC 11142, CECT 473), were supplied by the Colección Española de Cultivos Tipo (CECT). The pathogenic strain was grown in nutrient broth (NB, Sigma-Aldrich, St. Louis, MO, USA) and *Ax* in Hestrin-Schramm (HS) agar [28] at 30 °C. All HS medium constituents were purchased from Sigma-Aldrich. *Lactobacillus fermentum*, *Lf*, was kindly provided by Biosearch Life S.A. and grown in de Man, Rogosa and Sharpe medium (MRS, Oxoid, Hampshire, UK) at 37 °C.

### 2.4. Synthesis of Bacterial Cellulose

The synthesis of bacterial cellulose (BC) was carried out by culturing a single *Ax* colony, grown on agar culture medium, in 6 mL of HS medium. After 3 days of incubation at 30 °C, the BC pellicle was vortexed in order to remove active cells embedded in the membrane. One milliliter of the suspension was transferred to a 250 mL Erlenmeyer flask containing 100 mL of HS liquid medium, and incubated at 30 °C in static conditions for a week.

After incubation, BC pellicles produced on the liquid–air interface of each culture were harvested, cut into pieces of similar size and weighted. The pieces had a thickness of 3 mm and weighted 0.15 ± 0.002 g. The pellicles were then purified by immersing in EtOH, boiled in water for 40 min, immersed in NaOH 0.1 M at 90 °C for 1 h (with four dissolution replacements), and neutralized in distilled water. Finally, the pellicles were sterilized for 20 min in an autoclave at 121 °C.

### 2.5. Incorporation of Probiotics into BC

BC pieces were placed over MRS medium previously inoculated with 10^6^ colony forming units (CFU)/mL of lyophilized *Lf*, avoiding the complete immersion of samples, for 3 h at 37 °C.

### 2.6. Incorporation of Silver Nanoparticles

The side free of probiotics was immersed in the filtered-AgNPs solution for times ranging from 5 to 30 min. Afterwards, the pellicles were repeatedly rinsed with sterile ultrapure water in order to remove the excess of citrate, silver ions and non-adsorbed AgNPs. The biomaterial containing the highest AgNPs concentration with null effect on the probiotic viability was obtained by incubating BC-*Lf* for 15 min. This biomaterial was referred to as AgNP-BC-*Lf*.

### 2.7. Quantification of Immobilized Probiotics on BC

The protocol used to quantify the number of probiotics adsorbed on the BC pellicles was recently described [26]. Briefly, functionalized BC (0.15 g) was digested with cellulase from *Trichoderma reesei* (No C2730-50ML, Sigma–Aldrich). Then, probiotics were suspended in 5 mL of saline solution and colony-forming units (CFU) were determined by counting in MRS-agar plates after 24 h of incubation (37 °C) in anaerobic conditions (using the BD GasPak TM ES Anaerobe Container System, Hamilton, NJ, USA). The serial dilution with a number of visible colonies around 20–300 was used to calculate CFU, and plating was performed in triplicate.

### 2.8. Field Emission Scanning Electron Microscopy (FESEM)

Cellulosic samples were fixed in 1 mL of cacodylate buffer (0.1 M, pH 7.4) containing 2.5% of glutaraldehyde at 4 °C for 24 h. Subsequently, samples were washed with cacodylate buffer three times for 30 min at 4 °C. The samples were stained with osmium tetroxide (OsO_4_) solution (1% *v*/*v*) for 2 h in the dark, being then repeatedly rinsed with Milli-Q water to remove the excess of OsO_4_ solution. Samples were then dehydrated at room temperature with ethanol/water mixtures of 50%, 70%, 90% and 100% (*v*/*v*) for 20 min each, being the last concentration repeated three times and dried at the CO_2_ critical point. Finally, dehydrated samples were mounted on aluminum stubs using a carbon tape, sputtered with a thin carbon film, and analyzed using a FESEM (Zeiss SUPRA40V, Oberkochen, Germany) of the Centre for Scientific Instrumentation (University of Granada, CIC-UGR, Granada, Spain).

### 2.9. Bacterial Viability Assay

Viability of probiotics adhered to BC was assessed by confocal laser scanning microscopy (CLSM). The samples were washed with sterile water and stained with LIVE/DEAD BacLight Bacterial Viability Kit (ThermoFisher, Waltham, MA, USA) following manufacturer’s instructions. This assay combines membrane-impermeable DNA-binding stain, i.e., propidium iodide (PI), with membrane-permeable DNA-binding counterstain, SYTO9, to stain dead and live and dead bacteria, respectively. Cell viability along the BC matrix was evaluated with a confocal microscope (Nikon Eclipse Ti-E A1, Nikon, Tokio, Japan) of the CIC-UGR equipped with 20 × objective. For acquiring SYTO9 signals (green channel), a 488 nm laser and 505–550 nm emission filter was used. For PI (red channel), a 561 nm laser and 575 nm long-pass emission filter were used. Images were analyzed with NIS Elements software (Nikon, Tokio, Japan).

### 2.10. Antimicrobial Activity Studies: Agar-Diffusion Method

Antimicrobial activity of pure BC (as a negative control), BC with AgNPs (as a positive control), BC with *Lf* and BC with AgNPs and *Lf* against *PA* was assessed by the agar-diffusion method [29]. In brief, 0.1 mL of an overnight culture of *PA* was spread on nutrient agar Petri dish. Then, cellulosic samples were placed on the agar plate containing the pathogen and incubated 24 h at 37 °C before examination of inhibition zones.

Probiotic CFUs before and after every antibacterial test were determined as described above (quantification of immobilized probiotics on BC). Differences lower than 10% were observed.

Results were analyzed with the software GraphPad Prism 5, and data are expressed as mean ± standard deviation. For the statistical analysis, we applied the one-way ANOVA, Bonferroni’s method.

## 3. Conclusions

A new concept of bifunctional BC, based on the combination of two antibacterial agents, i.e., probiotics (*Lf*) and AgNPs was here developed. The antibacterials were intentionally placed on opposite faces to avoid the killer effect of AgNPs on the probiotics. The antibacterial assays against *PA* pointed out that the activity of AgNP-BC-*Lf* is the result of an additive effect of both antibacterial components. This activity against *PA*, together with the extraordinary features of BC as a wound dressing, makes AgNP-BC-*Lf* an antibacterial with potential applications in skin infections without the use of antibiotics.

## Figures and Tables

**Figure 1 molecules-26-02848-f001:**
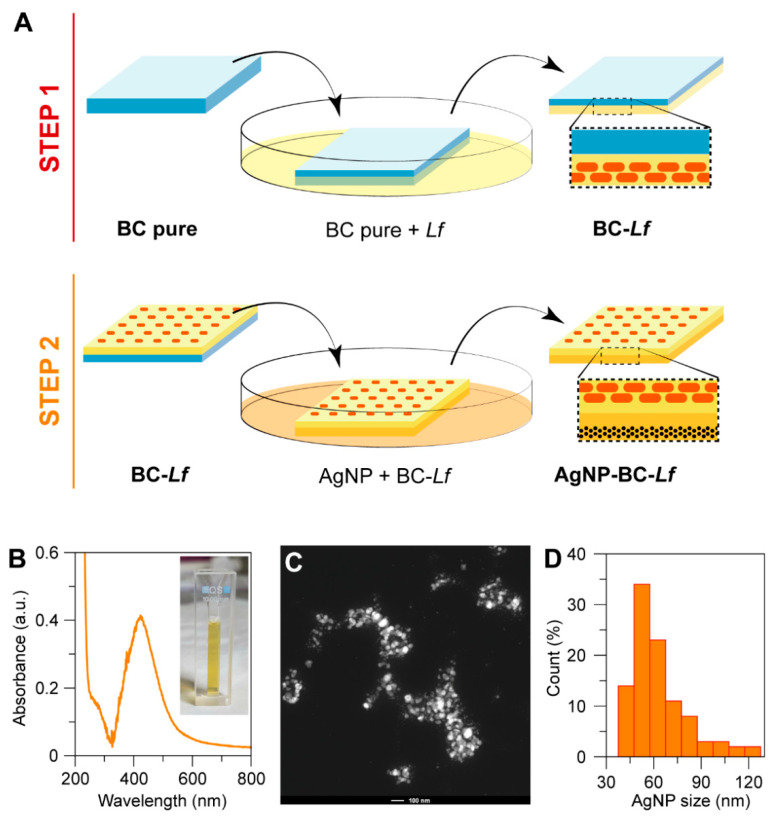
(**A**) Graphical representation of the experimental protocol used to obtain the two-sided BC. The first step involves the adsorption of probiotics followed by the impregnation of the opposite BC side with AgNPs. The resulting material is referred to as AgNPs-BC-*Lf*. (**B**) The UV-vis spectrum showed the expected absorbance band centered at 420 nm, in agreement with AgNPs with a mean diameter of 50 nm. The inset in B corresponds to a picture of an AgNPs solution. (**C**) HAADF-STEM micrograph of AgNPs (scale bar is 100 nm). (**D**) Diameter distribution of AgNPs (n = 100).

**Figure 2 molecules-26-02848-f002:**
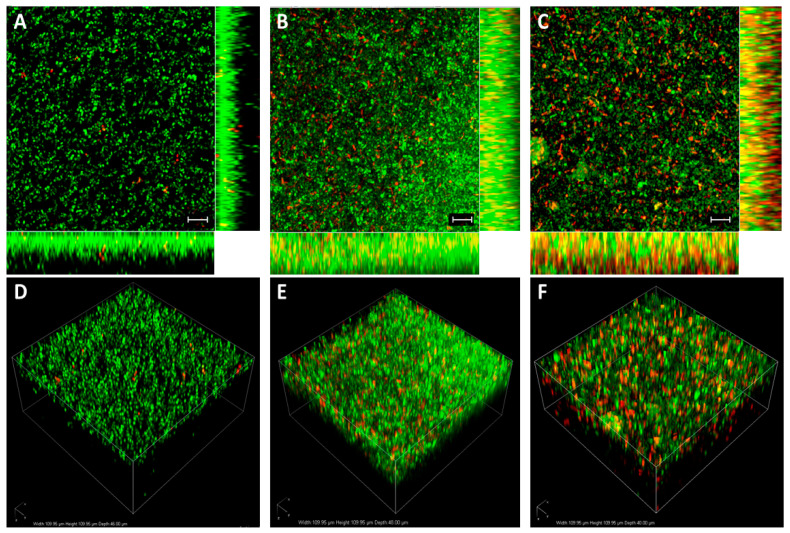
CLSM of BC-*Lf* at different incubation times in AgNPs solution. Samples were stained with SYTO9/PI dyes. The images are maximum intensity projections of 20 μm in-depth of (**A**) BC after *Lf* adsorption (BC-*Lf*), (**B**) BC-*Lf* after 15 min of incubation in AgNPs (AgNP-BC-*Lf*), and (**C**) BC-*Lf* after 30 min of incubation in AgNPs. Scale bars = 10 μm. (**D**–**F)** images are three-dimensional reconstructions of (**A**–**C**), respectively.

**Figure 3 molecules-26-02848-f003:**
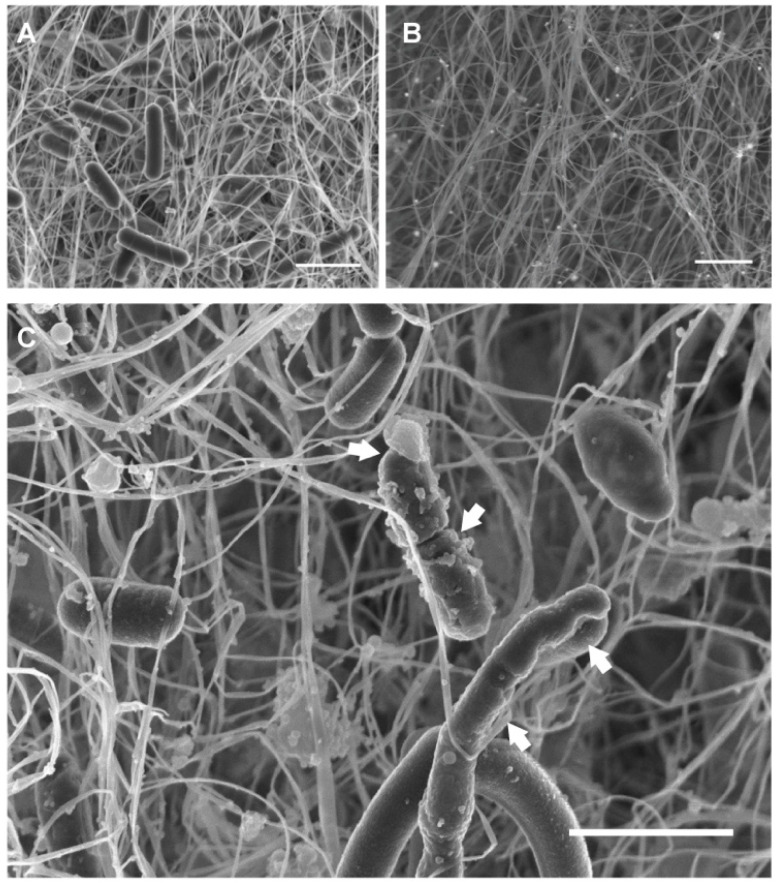
FESEM images of the two-sided biomaterials. (**A**) The side containing *Lf* exhibited the typical rod-like morphology of this bacterium. (**B**) BC side containing AgNPs. (**C**) *Lf*-functionalized BC surface, containing a higher amount of AgNPs (30 min of incubation in the AgNPs solution). Arrows show the cell wall damage. Scale bars: 2 μm.

**Figure 4 molecules-26-02848-f004:**
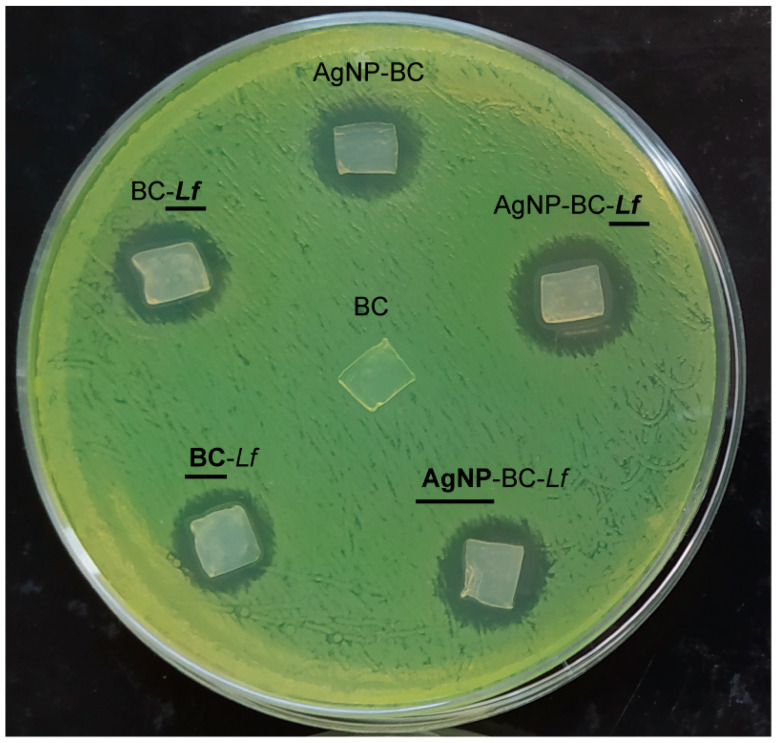
Inhibitory activity of the composite materials against *PA*. Inhibition zones of BC, AgNP-BC, BC-*Lf* and AgNP-BC-*Lf*. The corresponding BC side in contact with agar is marked in bold and underlined. The diameter of the Petri dish is 9 cm.

**Table 1 molecules-26-02848-t001:** Area of the inhibition zones around the cellulosic samples in the antimicrobial assays shown in Figure 4. Results are shown as means of three replicates ± standard deviation (SD). Statistical analysis was carried out by the one-way ANOVA (Bonferroni’s method). Letters indicate significant differences (*p* < 0.05) between samples. The BC side in direct contact with agar is indicated in bold and underline letters.

Muestra	Inhibition Zones ± SD (mm^2^)
BC	0 ± 0 a
AgNP-BC	89 ± 2 b
BC-***Lf***	81 ± 3 b
**BC**-*Lf*	80 ± 2 b
AgNP-BC-***Lf***	137 ± 9 c
**AgNP**-BC-*Lf*	110 ± 10 c

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
