# Peer review of "Two-Sided Antibacterial Cellulose Combining Probiotics and Silver Nanoparticles"

_molecules, 2021, doi:10.3390/molecules26102848_

Round 1
Reviewer 1 Report
This investigation is focused on obtaining of a hybrid material based on silver nanoparticles and living probiotic using bacterial cellulose as scaffold for antibacterial applications. Although this manuscript reports experimental results that should be interesting, there are concerns that the authors should address:
- It is recommend that the authors indicates the synthesis route that was followed for silver nanoparticles obtaining.
- If the silver nanoparticles were functionalized prior their incorporation to the bacterial cellulose scaffold, DLS is not an adequate technique to determinate particles diameter. Thus, it is recommended to perform direct measure of particle size by transmission electron microscopy techniques, such as HAADF-STEM.
- It is also recommended that authors includes detailed conditions for the incorporation of living probiotic and silver nanoparticles to the scaffold. How the authors measure the incorporation of these species to the scaffold? What is the concentration of these species in the scaffold?
Author Response
[Comment #1]: It is recommend that the authors indicates the synthesis route that was followed for silver nanoparticles obtaining.
[Reply]: Silver nanoparticles (AgNPs) were prepared by the well-known Turkevich method, using citrate as reducing and capping agent. The experimental details were already described in section 3 (Materials and Methods, Synthesis of silver nanoparticles). To avoid further misleading, we have included a sentence in the main text (page 4) indicating that the synthetic route was the Turkevich metod, as follow:
“AgNPs with average diameter of 50 nm (Figure 1B-C) were synthesized by the Turkevich method, using citrate as reducing and capping agent [27].”
[Comment #2]: If the silver nanoparticles were functionalized prior their incorporation to the bacterial cellulose scaffold, DLS is not an adequate technique to determinate particles diameter. Thus, it is recommended to perform direct measure of particle size by transmission electron microscopy techniques, such as HAADF-STEM.
[Reply]: According to the referee’s comment, we have determined the AgNPs diameter by electron microscopies (HAADF-STEM). Diameter distribution (histogram) from more than 100 nanoparticles has been added as Figure 1C. An appropriate sentence has been added in the main text (page 4) and the corresponding experimental details in the material and methods.
[Comment #3]: It is also recommended that authors includes detailed conditions for the incorporation of living probiotic and silver nanoparticles to the scaffold.
[Reply]: This information was already included in the original version of the manuscript. Two subsections of the materials and methods were dedicated to the experimental conditions used to incorporate probiotics (title: Incorporation of probiotics into BC) and AgNPs (title: Incorporation of silver nanoparticles) in page 7. Authors believe that these experimental details are enough to be reproduced by any person.
[Comment #4]: How the authors measure the incorporation of these species to the scaffold? What is the concentration of these species in the scaffold?
[Reply]: After the referee’s question we have measured the probiotic amount in the final biomaterial as follows: The biomaterials were treated with the cellulase enzyme until complete digestion. Then, probiotics were determined by counting CFUs in MRS-agar cultures.
A specific subsection, titled “Quantification of immobilized probiotics on BC”, describing the procedures has been added to materials and methods. Results have been also stated in the main text (page 4).
We detected the presence of AgNPs by electron microscopies but we did not determine their immobilization level. The presence of suspended probiotic bacteria made hard the determination of the silver concentration by ICP-OES.
Reviewer 2 Report
Paper entitled „Two-sided antibacterial cellulose combining probiotics and silver nanoparticles” reminded me what a pleasure it is to read well-written short communiques. There is exactly enough material in the paper to describe this new interesting setup and nothing more. The paper should be published as it is now.
Author Response
We warmly thank the reviewer for the positive comments on our manuscript.
Reviewer 3 Report
The work is very interesting and innovative. However, in my opinion, there are many methodological inaccuracies here.
BC is an extremely absorbent material because of its porosity. Therefore, it is often used as a carrier for the immobilization of bioactive substances with antimicrobial action and microorganisms with high biotechnological potential.
The AgNPs immobilization time at BC has been adjusted. How can we be sure that AgNPs particles did not penetrate to the part of BC where Lf was immobilized during agar-diffusion test? Lf viability should be checked after an agar-diffusion test.
How thick was BC?
What was the effectiveness of the Lf immobilization?
What was the final concentration of AgNPs in the biomaterial?
Line 115-125. Longer incubation time of BC with AgNPs, the higher the number of dead probiotics. The amount of AgNPs in BC increases with the time of incubation. Please explain how "This finding evidences that BC provides probiotics with extra protection".
Line 94-95. In my opinion, the described method of Lf immobilization (Incorporation of probiotics into BC) is an adsorption method, not an adsorption-incubation method.
The adsorption-incubation method is well explained in this paper
Żywicka, A., Wenelska, K., Junka, A. et al. Immobilization pattern of morphologically different microorganisms on bacterial cellulose membranes. World J Microbiol Biotechnol 35, 11 (2019). https://doi.org/10.1007/s11274-018-2584-7
Were the differences in agar-diffusion test obtained for AgNP-BC-Lf, AgNP-BC, and BC-Lf statistically significant? The results of the inhibition zone for individual samples should be provided in the table or in the graph along with the standard deviation and statistics.
Author Response
[Comment #1]: The work is very interesting and innovative. However, in my opinion, there are many methodological inaccuracies here.
[Reply]: We warmly thank the reviewer for the positive comments on our manuscript.
[Comment #2]: BC is an extremely absorbent material because of its porosity. Therefore, it is often used as a carrier for the immobilization of bioactive substances with antimicrobial action and microorganisms with high biotechnological potential. The AgNPs immobilization time at BC has been adjusted. How can we be sure that AgNPs particles did not penetrate to the part of BC where Lf was immobilized during agar-diffusion test? Lf viability should be checked after an agar-diffusion test.
[Reply]: The referee is right. We have determined the probiotic CFUs after the agar-diffusion tests and compared to those of the starting biomaterials. The CFUs determination was carried out by previous sample digestion with cellulase. The procedures and results are included in the experimental of the revised manuscript.
We found a certain decrease in the probiotic CFUs, not higher than 10%, after the agar-diffusion test.
We have included these results in the main text: “On the other hand, it is interesting to note that the antibacterial activity of AgNP-BC-Lf, no matter its face, is really a sum of the two antibacterial components (AgNPs and Lf). The possible silver cations diffusion inside the biomaterial during the agar-diffusion test, from AgNPs to Lf, and the consequent probiotic death resulted negligible. In fact, the probiotic viability before and after the antibacterial test assays was close, with CFUs decreases lower than 10% (see Materials and methods)”
[Comment #3]: How thick was BC?
[Reply]: The thickness of BC, 3 mm, was stated in the materials and methods section of the revised manuscript.
[Comment #4]: What was the effectiveness of the Lf immobilization? What was the final concentration of AgNPs in the biomaterial?
[Reply]: As discussed in the comment #4 of reviewer #1, we have measured the probiotic concentrations in the final biomaterial as follows: The biomaterials were treated with the cellulase enzyme until complete digestion. Then, probiotics were determined by counting CFUs in MRS-agar cultures.
A detailed description of procedures and results are included in the experimental (“Quantification of immobilized probiotics on BC”) of the revised manuscript. Results have been also stated in the main text (page 4).
[Comment #6]: Line 115-125. Longer incubation time of BC with AgNPs, the higher the number of dead probiotics. The amount of AgNPs in BC increases with the time of incubation. Please explain how "This finding evidences that BC provides probiotics with extra protection".
[Reply]: To avoid any confusion we have eliminated the sentence “This finding evidences that BC provides probiotics with extra protection”.
[Comment #7]: Line 94-95. In my opinion, the described method of Lf immobilization (Incorporation of probiotics into BC) is an adsorption method, not an adsorption-incubation method. The adsorption-incubation method is well explained in this paper Żywicka, A., Wenelska, K., Junka, A. et al. Immobilization pattern of morphologically different microorganisms on bacterial cellulose membranes. World J Microbiol Biotechnol 35, 11 (2019). https://doi.org/10.1007/s11274-018-2584-7
[Reply]: We have rephrased “probiotics and AgNPs were sequentially incorporated on opposite faces of BC by the adsorption-incubation methodology” as “probiotics and AgNPs were sequentially adsorbed on opposite faces of BC” in the main text.
[Comment #8]: Were the differences in agar-diffusion test obtained for AgNP-BC-Lf, AgNP-BC, and BC-Lf statistically significant? The results of the inhibition zone for individual samples should be provided in the table or in the graph along with the standard deviation and statistics.
[Reply]: We have added a table in the main text including the area of the inhibition zones with the corresponding standard deviations and statistics.
Round 2
Reviewer 1 Report
This investigation is focused on obtaining of a hybrid material based on silver nanoparticles and living probiotic using bacterial cellulose as scaffold for antibacterial applications. The revised version of manuscript includes significant improvements and consider the recommendations given to the authors. In addition, it is recommended to include and example of the HAADF-STEM images used to determinate the particle size distribution in Figure 1.
Author Response
Silver nanoparticles (AgNPs) were prepared by the well-known Turkevich method (ref 27). Size distributions of AgNPs obtained with this protocol are repeatedly reported in the literature. In this work, we initially used dynamic light scattering (DLS) to confirm that the size of the particles was in agreement to that previously reported (ca. 50 nm in average). Following the reviewer recommendation, we also determined the size distribution from particles imaged by TEM (HAADF-STEM). A representative HAADF-STEM image (Figure 1C) and the corresponding histogram (Figure 1D) has been included in this new version. We found particles with an average diameter of 50 nm, in agreement with that reported in reference 27 and many others.
Reviewer 3 Report
I have no additional questions.
Author Response
A HAADF-STEM image of AgNPs (Figure 1C) and the corresponding histogram (Figure 1D) have included in this new version. The figure caption has been modified accordingly.
We have also included a more detailed description of AgNPs size in the Experimental